# Oxytocin-mediated social enrichment promotes longer telomeres and novelty seeking

Jamshid Faraji[1,2], Mitra Karimi[3], Nabiollah Soltanpour[4], Alireza Moharrerie[5], Zahra Rouhzadeh[6], Hamid lotfi[7], S Abedin Hosseini[2], S Yaghoob Jafari[2], Shabnam Roudaki[8], Reza Moeeini[8]*, Gerlinde AS Metz[1]*

[1]Canadian Centre for Behavioural Neuroscience, University of Lethbridge, Lethbridge, Canada; [2]Faculty of Nursing & Midwifery, Golestan University of Medical Sciences, Gorgan, Iran; [3]Inclusive-Integrated Education Program for Children with Special Needs, Exceptional Education Organization, Tehran, Iran; [4]Department of Anatomical Sciences, Babol University of Medical Sciences, Babol, Iran; [5]Department of Anatomy, Golestan University of Medical Sciences, Gorgan, Iran; [6]Department of Psychology, Islamic Azad University, Sari, Iran; [7]Department of Psychology, Islamic Azad University, Tonekabon, Iran; [8]Department of Behavioural Studies, Avicenna Institute of Neuroscience, Yazd, Iran

**\*For correspondence:**
reza.moeeni123@gmail.com (RM); gerlinde.metz@uleth.ca (GASM)

**Competing interests:** The authors declare that no competing interests exist.

**Abstract** The quality of social relationships is a powerful determinant of lifetime health. Here, we explored the impact of social experiences on circulating oxytocin (OT) concentration, telomere length (TL), and novelty-seeking behaviour in male and female rats. Prolonged social housing raised circulating OT levels in both sexes while elongating TL only in females. Novelty-seeking behaviour in females was more responsive to social housing and increased OT levels than males. The OT antagonist (OT ANT) L-366,509 blocked the benefits of social housing in all conditions along with female-specific TL erosion and novelty-seeking deficit. Thus, females seem more susceptible than males to genetic and behavioural changes when the secretion of endogenous OT in response to social life is interrupted. Social enrichment may, therefore, provide a therapeutic avenue to promote stress resiliency and chances of healthy aging across generations.
DOI: https://doi.org/10.7554/eLife.40262.001

## Introduction

The quality and quantity of social relationships in the modern era are rapidly changing, with increasing social isolation, and many individuals no longer living in extended families or missing close confidants. Restricted social relationships can endanger public health and increase mortality rate (*Mikosz et al., 2016*). Males and females, both human and animal, respond differently to social interactions. Differences may range from changes in neurobiological processes (*Mikosz et al., 2016*; *Stack et al., 2010*; *Ngun et al., 2011*; *Cherif et al., 2003*) to displaying distinctive gender-specific behaviours (*Szell and Thurner, 2013*; *Jukka-Pekka Onnela et al., 2014*; *Palchykov et al., 2012*; *Trainor et al., 2011*). Genetic and hormonal determinants regulate social behaviours (*Choleris et al., 2018*) and dictate morphological and behavioural variations in males and females. Robust sexual dimorphism was found in the neuropeptide oxytocin (OT) receptor density throughout the rat brain (*Smith et al., 2017*) and in OT receptor signaling in the medial prefrontal cortex (mPFC), which may modulate social interactions in females in response to OT (*Nakajima et al., 2014*). Primarily synthesized in the paraventricular nucleus (PVN) and supraoptic nucleus of the

hypothalamus (*Jurek and Neumann, 2018*), OT is a key hormonal correlate of social behaviours in mammals (*Anacker and Beery, 2013*; *Knobloch and Grinevich, 2014*). OT exposure promotes social behaviours (*Gamer et al., 2010*; *Guastella et al., 2008*; *Andari et al., 2010*; *Lim and Young, 2006*; *Bernaerts et al., 2017*; *McGraw and Young, 2010*) and facilitates adaptive responses to stressors and stress resiliency (*Olff et al., 2013*). OT has also been shown to prevent premature aging of muscle tissue and facilitates muscle regeneration (*Elabd et al., 2014*). Hence, it can be hypothesized that OT exposure influences the chances of healthy aging.

A specific cellular index which differs according to age and sex indicates changes in telomere length (TL). Telomeres are DNA-repetitive nucleotide segments at the ends of each chromosome in mammals that protect genetic material from degradation during somatic cell division. Telomeric DNA in humans naturally shortens with age, and therefore TL represents a sensitive indicator of tissue-specific cellular aging (*Puterman et al., 2016*). As TL decreases along with proliferation in each cell division, telomere shortening can be considered a measure of biological aging (*Sahin and DePinho, 2010*; *Blackburn, 2005*). A variety of influences such as distress (*Epel et al., 2004*; *Epel et al., 2010*), childhood adversity (*Puterman et al., 2016*), interpersonal instability (*Drury et al., 2014*), social disadvantages and deprivation (*Theall et al., 2013*), and social isolation (*Aydinonat et al., 2014*) have been shown to be correlated with TL erosion. TL has also been shown to be positively associated with psychosocial factors, such as conscientiousness (*Edmonds et al., 2015*), optimism (*Schutte et al., 2016*), social support (*Uchino et al., 2012*), and social-pair mate choice (*Johnsen et al., 2017*).

It appears that females generally have longer telomeres than males (*Gardner et al., 2014*; *Barrett and Richardson, 2011*), and shorter telomeres in males are correlated with reduced social support (*Zalli et al., 2014*). This has led to the speculation that social experiences possibly make females more susceptible to telomere elongation than males, a lifestyle-dependent process that, in concert with sex-specific hormonal correlates results in longevity gender gaps (*Merrill et al., 2017*). Also, females show facilitated social learning compared to males (*Ervin et al., 2015*), along with estrogenic control of OT and OT receptor activity (*Anacker and Beery, 2013*). Accordingly, we have recently shown that prolonged social experiences change brain structure and behaviours in a sexually dimorphic manner in rats (*Faraji et al., 2018*).

Here, we investigated whether social experiences modulate genetic and behavioural repertoires through OT in male and female rats. Whereas OT facilitates novelty-seeking behaviour and TL, the OT antagonist (OT ANT) L-366,509 in socially raised male and female rats intermittently reduced OT secretion. The results suggest that extended social experiences modulate novelty seeking and TL through OT in a sex-dependent manner, with females being more susceptible than males.

## Results

### Experiment 1

### Social experience increased plasma OT concentration in females

*Figure 1A* shows changes in plasma OT concentrations across different groups as a function of housing condition (standard, *males*: n = 23, *females*: n = 21; social, *males*: n = 22, *females*: n = 22). A significant effect of housing condition was observed in terms of circulating OT concentration (social vs. standard; $F_{1,86}$=24.85, $p \leq 0.001$) and Group (social males and females vs. standard males and females; $F_{3,84}$=13.69, $p \leq 0.001$), suggesting that long-term social experiences had a significant impact on plasma OT in socially raised animals (98.82 ± 3.94 vs. 71.00 ± 3.94 pmol/L). Post-hoc Tukey comparisons also indicated that social females showed higher OT concentration than standard females (108.55 ± 5.25 vs. 80.65 ± 5.37 pmol/L; $p \leq 0.002$). Furthermore, a marginal difference was observed between social males and females (89.09 ± 5.25 vs. 108.55 ± 5.25 pmol/L; $p = 0.05$; Post-hoc Tukey). No effect was found in terms of litter.

### Social experience increased telomere length (TL) in females but not males

Examination of TL in ear notch skin cells showed that socially raised rats (males: n = 31, females: n = 23) had greater telomere length than standard animals (males: n = 28, females: n = 27; $F_{1,107} = 24.91$, $p \leq 0.001$). Also, a significant effect of group was observed ($F_{3,105} = 65.84$, $p \leq 0.001$) while social female rats displayed greater TL than standard rats and their social male

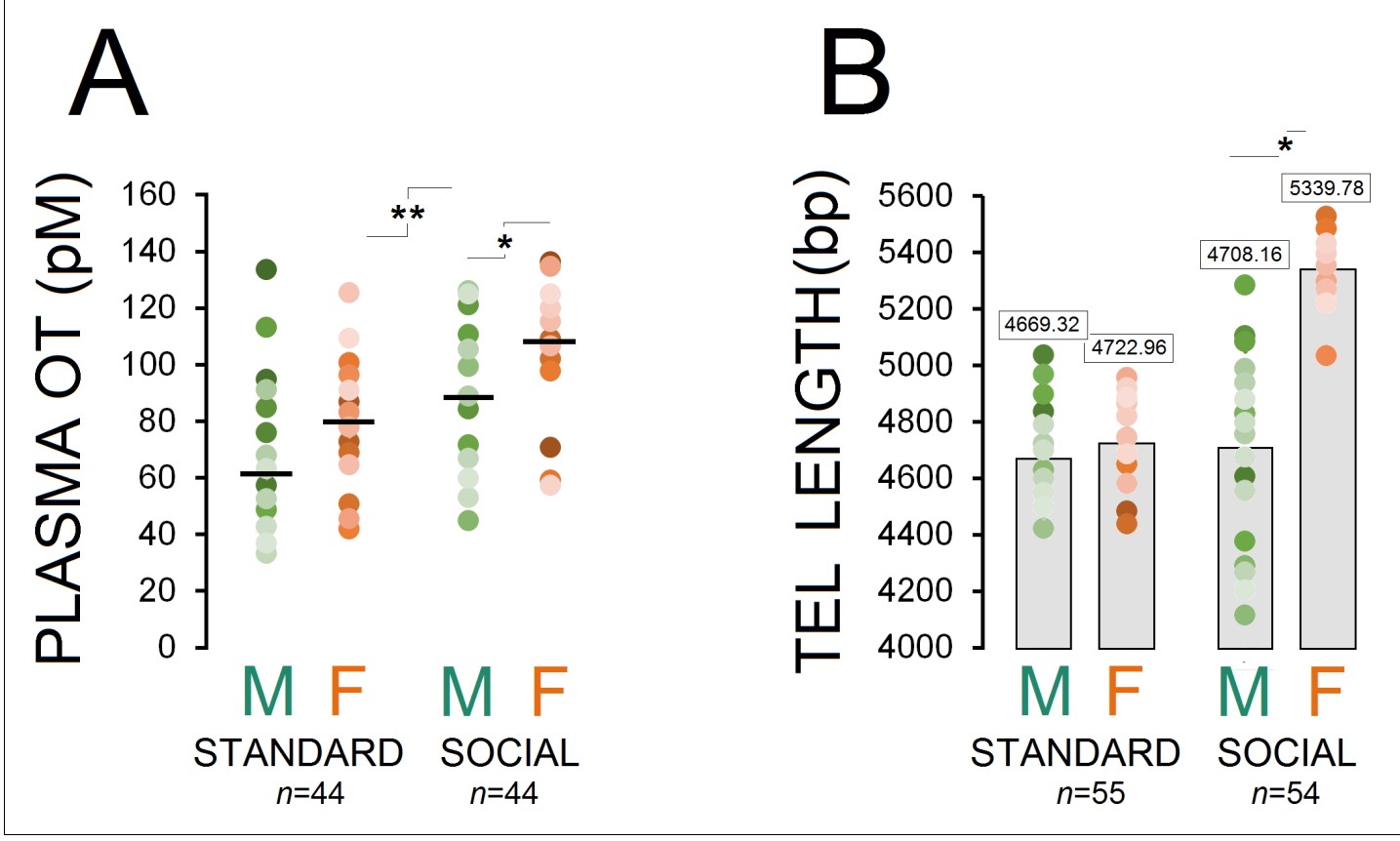

**Figure 1.** Housing conditions influence plasma OT and telomere length. (A) Social enrichment altered plasma OT levels in socially raised animals. Social females showed higher OT concentration than social males (n = 21–23/group). (B) Social experience increased telomere length only in females (n = 23) when compared with standard rats (n = 27–28) and their social male counterparts (n = 31). Asterisks indicate significant differences: *p ≤ 0.05; **p ≤ 0.01; one-way ANOVA. Filled circles: mean OT concentrations in individual rats. Horizontal bars: mean OT concentrations in each group. pM: picoMolar, OT: oxytocin, Tel: telomere, bp: basepair.

DOI: https://doi.org/10.7554/eLife.40262.002

The following source data is available for figure 1:

**Source data 1.** Housing conditions influence plasma OT and telomere length.
DOI: https://doi.org/10.7554/eLife.40262.003
**Source data 2.** Housing conditions influence plasma OT and telomere length.
DOI: https://doi.org/10.7554/eLife.40262.004

counterparts (all p ≤ 0.001; Post-hoc Tukey). No difference was found between social males and standard males and females (all p ≥ 0.05; Post-hoc Tukey; *Figure 1B*). No effect was found in terms of litter.

## Novelty-seeking behaviour in females was significantly influenced by social experiences

An illustration of the corridor field task (CFT) with and without central object along with paths taken by rats in standard and social groups is shown in *Figure 2A and B*. No-central object CFT: panel (C) compares the exploratory behaviour in both groups (standard: n = 33; social: n = 39) within different zones of the no-central object CFT. There was a significant main effect of housing condition (two levels) in terms of the time spent in each zone (corridor: $F_{1,70} = 39.60$, p ≤ 0.001; open: $F_{1,70} = 20.88$, p ≤ 0.001; central: $F_{1,70} = 16.43$, p ≤ 0.001). When compared with the standard group, social rats spent less time in the corridor zone (243.02 ± 8.8 s vs. 325.60 ± 9.6 s) and more time in the open (177.76 ± 7.9 s vs. 123.84 ± 8.6 s) and central zones (59.20 ± 4.7 s vs. 30.54 ± 5.2 s). No effect was found in terms of litter, and no significant interactions between factors were found, except for

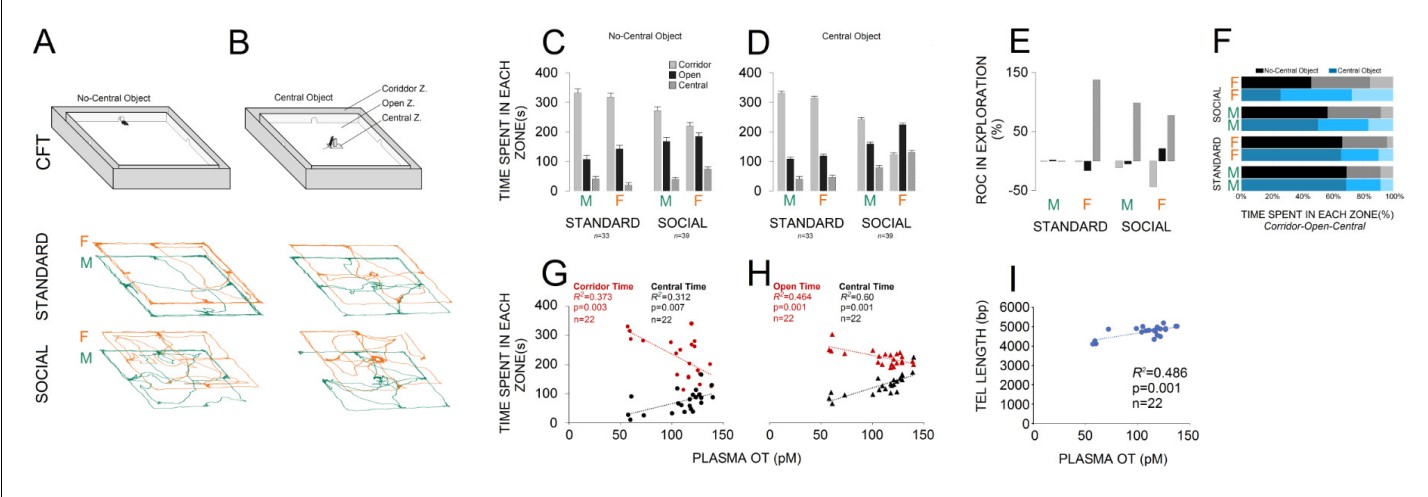

**Figure 2.** Social experience alters novelty-seeking behaviour in the corridor field task (CFT). (**A** and **B**) Illustration of the no-central and central-object CFT protocol along with samples of paths taken by rats in standard and social groups. (**C** and **D**) Social life affected novelty-seeking behaviour in a sexually dimorphic manner. Socially reared females (n = 22) explored the open and central zones more than their social male counterparts (n = 17) or standard group (n = 16 and 17). (**E** and **F**) The rate of changes (ROC) indicated the most profound impact on social females than any other group from no-central object to central-object versions of the CFT in the open and central zones. Also, a significant regression equation was found only in social females (n = 22) where OT concentration significantly predicts the exploration in the corridor and central zones (**G**) when central object was not presented. (**H**) Analysis of linear regression indicated significant regression equations for the time spent in open and central zones only in social females (n = 22) by which the increased plasma OT levels significantly predicted exploration time in CFT when the central object was presented. (**I**) The TL elongation, also, was significantly associated with an increase in plasma OT level only in social females (n = 22).

DOI: https://doi.org/10.7554/eLife.40262.005

The following source data is available for figure 2:

**Source data 1.** Social experience alters novelty-seeking behaviour.

DOI: https://doi.org/10.7554/eLife.40262.006

sex × litter (p ≤ 0.03). Furthermore, between-subjects analysis indicated a significant effect of Group (four levels) for all zones (three levels; all p = 0.000) showing that social females spent significantly less times in the corridor zone than any other group (all p ≤ 0.02; Post-hoc Tukey) and made more visits to the open zone than standard rats (all p ≤ 0.05; Post-hoc Tukey). No significant difference was found between social males and females in open zone exploration (p ≤ 0.6; Post-hoc Tukey). Social females also spent more time in the central zone when compared with standard housed animals and their male counterparts (all p ≤ 0.01; Post-hoc Tukey).

Central-object CFT: A significant effect of housing condition was observed in terms of the time spent in corridor ($F_{1,70}$ = 154.88, p ≤ 0.001), open ($F_{1,70}$ = 112.75, p ≤ 0.001) and central zones ($F_{1,70}$ = 65.65, p ≤ 0.001) of the CFT with central object (*Figure 2D*). Again, social animals explored the corridor zone less than standard rats (175.23 ± 8 s vs. 322.93 ± 8.7 s) and spent more exploration time in the open (196.23 ± 5.27 s vs. 113.51 ± 5.7 s) and central zones (108.53 ± 5.4 s vs. 43.54 ± 5.9 s) when compared with standard rats. Also, a significant effect of group was found for the corridor ($F_{3,68}$ = 115.42, p ≤ 0.001), open ($F_{3,68}$ = 42.31, p ≤ 0.001), and central zones ($F_{3,68}$ = 17.01, p ≤ 0.001), indicating that social females explored the corridor zone less and other zones more than any other group, even their male counterparts (all p ≤ 0.001; Post-hoc Tukey). No effect was found in terms of litter or interaction between factors (all p ≥ 0.05). Thus, social experience appeared to affect novelty-seeking explorative behaviours in the CFT in a sexually dimorphic manner, by which socially reared females explored the open and central zones more than social males and standard females. This behavioural flexibility was also supported by additional ROC analysis showing that social females experienced greater changes in novelty-seeking behaviours than other groups from no-central object to central-object versions of the CFT in the open and central zones (*Figure 2E and F*).

Simple linear regression analyses were conducted to predict novelty-seeking behaviour and TL based on plasma OT concentration. Plasma OT concentration and exploratory behaviour in no-

central object CFT: a significant regression equation was found ($F_{1,20}$ = 11.92, p ≤ 0.003) only in social females (n = 22) with an $R^2$ = 0.373, through which the OT concentration significantly predicts the exploration in the corridor zone (420.34 + −1.84). The observed regression equation for exploration in the central zone was also significant ($F_{1,20}$ = 9.08, p ≤ 0.007; $R^2$ = 0.312; *Figure 2G*). Therefore, only in social females does plasma OT concentration predict a particular profile of CFT exploration through which in the presence of enhanced OT concentration, the corridor zone was less explored whereas the central zone was more explored in social females when compared to other groups.

Plasma OT concentration and exploratory behaviour in central object CFT: analysis of linear regression indicated significant regression equations for the time spent in open ($F_{1,20}$ = 17.33, p ≤ 0.000; $R^2$ = 0.464) and central ($F_{1,20}$ = 30.01, p ≤ 0.000; $R^2$ = 0.60) zones by which the increased plasma OT levels only in social females significantly predicted exploration time (*Figure 2H*). No significant changes were observed for the time spent in the corridor zone (p = 0.07; $R^2$ = 0.148).

Plasma OT concentration and TL: a significant regression equation was found ($F_{1,20}$ = 18.88, p ≤ 0.000; $R^2$ = 0.486) only in social females through which the predicted TL was equal to 3816.88 + 8.53 (OT concentration) bp when OT concentration was measured in pM (*Figure 2I*). Thus, TL elongation was associated with higher plasma OT level only in social females.

## Experiment 2

### The OT antagonist L-366,509 reduced circulating plasma OT levels

Circulating OT concentration showed a significant effect of group (n = 10–13/group, $F_{7,83}$ = 14.64, p ≤ 0.001), whereas injection of the OT antagonist L-366,509 resulted in a significant decrease in all ANT groups (all p ≤ 0.05, *Post-hoc* Tukey) except for standard OT ANT males (p = 0.475, *Post-hoc* Tukey; *Figure 3A and B*). Also, a significant effect of housing condition (n = 45 and 46, $F_{1,89}$ = 9.72, p ≤ 0.002) was observed suggesting that social rats still had greater concentration of the plasma OT when compared with standard animals (80.55 ± 2.72 vs. 68.43±2.75 pmol/L). A significant effect of sex (n = 44 and 47, $F_{1,89}$ = 11.08, p ≤ 0.001) indicated greater levels of OT in females than males (81.17 ± 2.76 vs. 68.36±2.67 pmol/L).

Social control females displayed higher plasma OT concentration than any other group (all p ≤ 0.05, *Post-hoc* Tukey) except for social control males (p = 0.07, *Post-hoc* Tukey). There were also significant differences in plasma OT with social control males having significantly higher OT concentration than other groups (p ≤ 0.05, *Post-hoc* Tukey) except for standard control females and social OT ANT females (all p ≥ 0.05, *Post-hoc* Tukey). A significant effect of litter was also found in terms of OT concentration ($F_{10,81}$ = 11.46, p ≤ 0.03), in which litters 4 and 7 displayed higher OT levels than other litters (all p ≤ 0.05, *Post-hoc* Tukey). Thus, the OT antagonist administration had a significant impact on the plasma OT concentration in OT ANT groups, particularly in females regardless of their housing condition.

### OT antagonist in social females significantly reduced telomere length

Despite significant effects of housing condition (n = 48 and 45/group, $F_{1,91}$ = 4.80, p ≤ 0.03), sex (n = 47 and 46/group, $F_{1,91}$ = 14.63, p ≤ 0.001), and group (n = 10–13/group, $F_{7,85}$ = 7.06, p ≤ 0.001), the administration of the OT antagonist L-366,509 had no significant effect on the TL in OT ANT groups, except for social OT ANT females (*Figure 3C and D*). Only social females had greater TL than other groups (all p ≤ 0.05, Post-hoc Tukey), replicating the results of Experiment 1. In turn, the OT antagonist abolished the benefits of social housing by reducing circulating OT and reducing TL in females only. No effect of litter was found.

### OT antagonist significantly influenced novelty-seeking behaviour in all groups

*Figure 4A and C* compare novelty-seeking behaviour in the CFT in animals with and without OT antagonist L-366,509 treatment. No-central object CFT: administration of L-366,509 affected novelty-seeking behaviour within all zones of the CFT across groups. A significant effect of housing condition ($F_{1,53}$ = 11.63, p ≤ 0.001) indicated that socially raised rats (n = 27) still explored the corridor zone less than standard animals (n = 28; 276.77 ± 8.99 s vs. 319.75 ± 8.82 s). However, when compared with standard rats, social rats explored open and central zones of the CFT more frequently

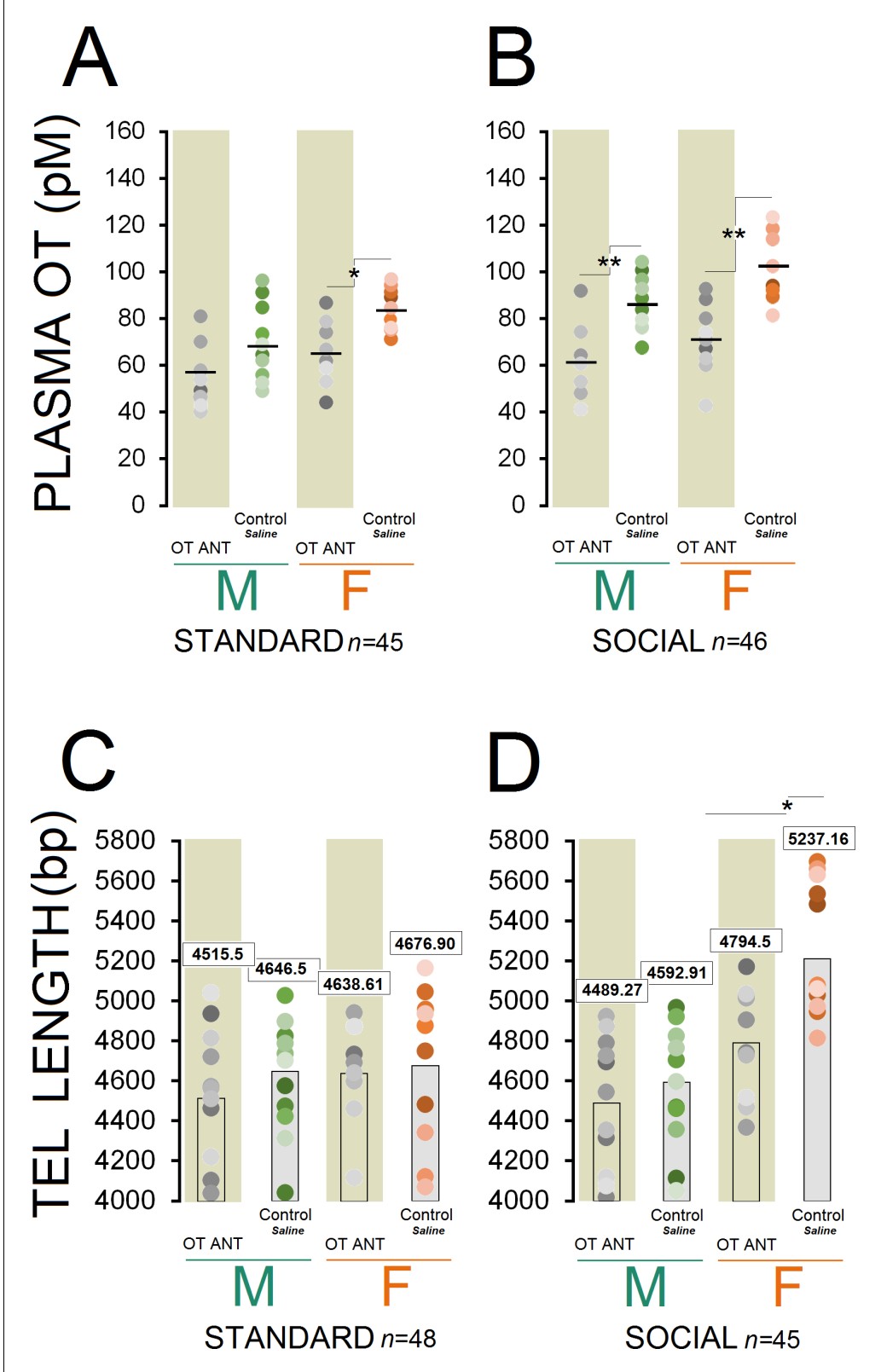

**Figure 3.** OT antagonist L-366,509 blocks the social experience phenotype in plasma OT concentration and telomere length. (**A** and **B**) OT antagonist administration reduced plasma OT concentration in all OT ANT groups, except for standard OT ANT males (n = 10–13/group). Social control females displayed higher concentration of plasma OT than any other group. No difference was found between social control females and males. (**C** and **D**) OT antagonist L-366,509 reduced telomere length in social females. Asterisks indicate significant differences: *p ≤ 0.05; **p ≤ 0.01; one-way ANOVA. Filled
*Figure 3 continued on next page*

*Figure 3 continued*

circles: mean OT concentrations in individual rats. Horizontal bars: mean OT concentrations in each group. ANT: antagonist, bp: basepair, OT: oxytocin, pM: picoMolar, Tel: telomere.

DOI: https://doi.org/10.7554/eLife.40262.007

The following source data is available for figure 3:

**Source data 1.** OT antagonist blocks phenotype.
DOI: https://doi.org/10.7554/eLife.40262.008

**Source data 2.** OT antagonost blocks social experience phenotype.
DOI: https://doi.org/10.7554/eLife.40262.009

(open: 153.66 ± 6.48 s vs. 119.10 ± 6.3 s; central: 49.55 ± 5.19 s vs. 41.14 ± 5.10 s). The group effect was also significant for all three zones of the CFT (all p ≤ 0.01). Only social rats, however, were significantly impacted by the OT antagonist in terms of exploratory behaviour. Essentially, social control females significantly explored the central zone more than standard control females, whereas their corridor time diminished (all p ≤ 0.01, Post-hoc Tukey). Social control females also spent less time in the corridor and more time in the open and central zones when compared with standard OT ANT females (all p ≤ 0.01, Post-hoc Tukey). Moreover, when compared with social OT ANT females, the social control females explored the corridor zone less and spent more time in the central zone (all

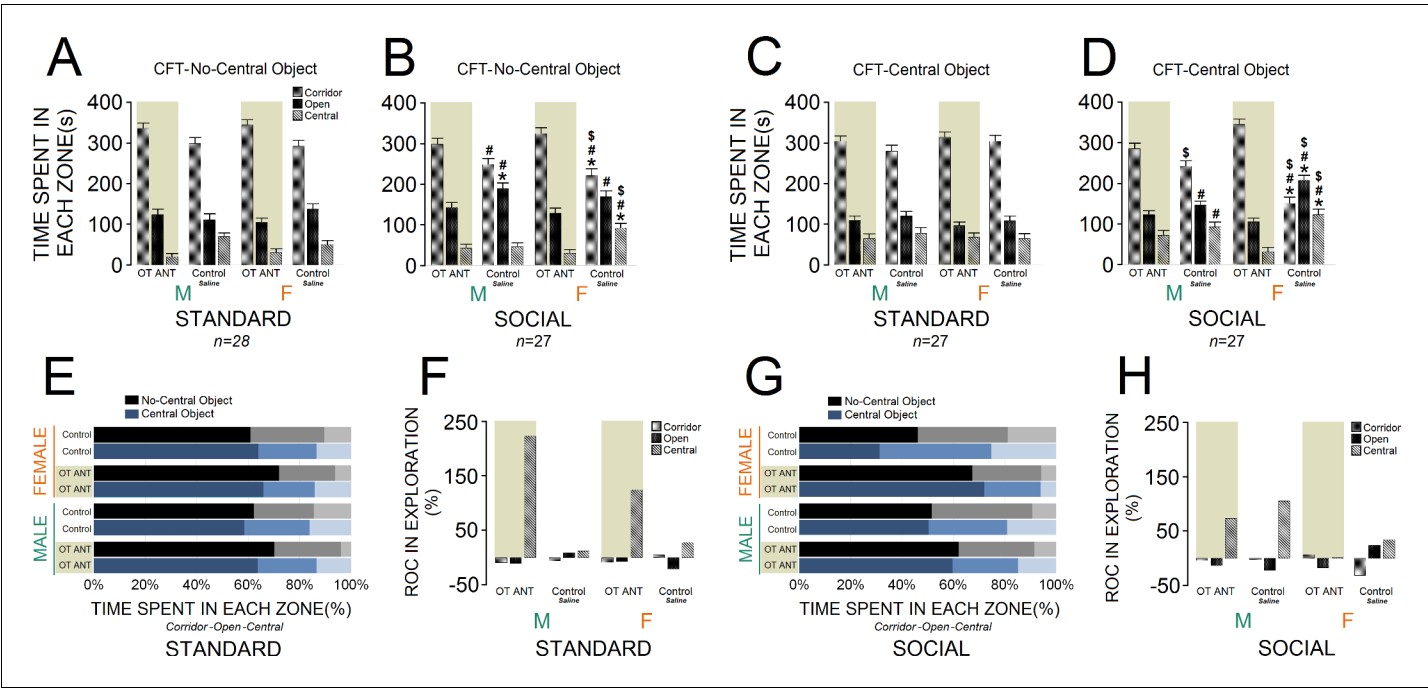

**Figure 4.** Behavioural consequences of the administration of OT antagonist L-366,509 in the corridor field task (CFT). (**A** and **B**) The OT antagonist affected novelty-seeking behaviour in all zones of the no-central object CFT (standard: n = 28; social: n = 27). (**C** and **D**) The OT antagonist had a significant impact on the social females' exploration in the central-object CFT. Novelty-seeking behaviour in socially raised females was more influenced by reduced OT levels than any other group (standard: n = 27; social: n = 27). (**E–H**) Rate of changes (ROC) within no-central and central-object CFT in response to OT antagonist. Note that social OT ANT females experienced the fewest changes in novelty-seeking behaviour from no-central object to central-object versions of the CFT. Asterisk indicates significant differences: p ≤ 0.05; MANOVA. Symbols denote comparisons: social control females: * relative to standard control females, # relative to standard OT ANT females, $ relative to social OT ANT females; social control males: * relative to standard control males, # relative to standard OT ANT males. Error bars show ± SEM. OT: oxytocin, ANT: antagonist.

DOI: https://doi.org/10.7554/eLife.40262.010

The following source data is available for figure 4:

**Source data 1.** Behavioural consequences of OT antagonist.
DOI: https://doi.org/10.7554/eLife.40262.011

p $\leq$ 0.05, Post-hoc Tukey). Thus, the OT antagonist L-366,509 changed the exploration pattern particularly in social females. There was no difference between social control males and females in terms of CFT exploration (all p $\geq$ 0.05, Post-hoc Tukey). When compared to standard control males, however, the social control males spent more time in the open zone (p $\leq$ 0.05, Post-hoc Tukey). The social control males also spent less time in the corridor zone but explored the open zone more than standard OT ANT males (all p $\leq$ 0.05, Post-hoc Tukey). No difference was observed between social control and social OT ANT males (all p $\geq$ 0.05, Post-hoc Tukey). All interactions between factors were insignificant, except for sex $\times$ litter and sex $\times$ litter $\times$ housing condition (all p $\leq$ 0.05). Thus, although exploration in the no-central object CFT version was somehow affected by L-366,509, CFT exploration in social females was more impacted by the OT antagonist than any other group.

Central-object CFT: *Figure 4C and D* illustrates novelty-seeking behaviour within the CFT with and without OT antagonist L-366,509. Again, a significant effect of housing condition indicated a different profile of novelty-seeking behaviour in social rats (n = 27). Socially raised rats explored the corridor zone less than standard animals (n = 27, $F_{1,52}$ = 5.45, p $\leq$ 0.02). Social animals, however, explored the open zone more than standard rats ($F_{1,52}$ = 10.97, p $\leq$ 0.002), whereas they showed no difference in the central zone (p $\geq$ 0.05). Also, significant effects of group for all zones (all p $\leq$ 0.05) showed that the OT antagonist L-366,509 had a greater impact on social female exploration (all p $\leq$ 0.05, Post-hoc Tukey). The inhibitory effect of L-366,509 on OT concentration, and consequently the novelty-seeking behaviour in socially raised males did not reach the levels of social females (all p $\leq$ 0.05, Post-hoc Tukey). No effect of litter, and no interactions between factors were found, except for litter $\times$ sex (p $\leq$ 0.04). Therefore, novelty-seeking behaviour in socially raised females was significantly more influenced by reduced OT levels than any other group. Further analysis of exploratory behaviour through the ROC also indicated that L-366,509 had greater impact on the novelty-seeking behaviour in social females in both no-central and central-object CFT (*Figure 4E–H*).

## Discussion

The present rodent study indicates that persistent social experience is causally associated with increased OT level, elevated exploratory behaviour, and telomere elongation. These changes occurred in a sex-dependent manner to support three main conclusions: (i) social experience increases plasma OT levels, (ii) enhanced novelty-seeking behaviour and TL in socially housed females is mediated by higher OT levels, and (iii) social females respond to interrupted endogenous OT secretion with TL erosion and novelty-seeking deficit. Thus, OT is arguably involved in mediating beneficial effects of social experience on behaviour and TL. The findings reveal a dynamic interplay between social interactions and OT hormone to promote genetic and behavioural resiliency in female rats (*Figure 5*).

### Prolonged social experiences increase OT in a sex-specific manner

The pivotal role of OT in social behaviour has become a target of many experimental efforts (*Churchland and Winkielman, 2012*). However, the majority of data linking OT to social behaviours represent correlational conclusions (*Barraza and Zak, 2009*), or depict a hormone-to-behaviour approach in which subjects were exposed to OT, and then were monitored for OT-induced behavioural alterations (*Bernaerts et al., 2017*). Using a causal, behaviour-to-hormone approach in the present experiments, we examined the direct impact of prolonged social interactions across development upon OT-mediated effects. In previous discussions (*Cerulo, 2009*; *Schilbach et al., 2013*; *Hari et al., 2015*) social interaction was regarded as a critical contributor to health and longevity (*Umberson and Montez, 2010*; *Yang et al., 2016*) and it was assumed that OT is involved in social behaviours and social memory (*Gamer et al., 2010*; *Bernaerts et al., 2017*; *Kirsch et al., 2005*; *Baumgartner et al., 2008*; *Kosfeld et al., 2005*; *Auyeung et al., 2015*). The present findings expand these earlier studies by demonstrating that social rearing elevates plasma OT concentration, and this hormonal response to social experience appears more prominent in females than males. Experiment 2 replicated these findings from Experiment 1.

There are two possible mechanisms that may underlie the dynamic interaction between social experience and OT. First, social interaction is associated with persistent sensory stimulation during development which may promote hypothalamic-pituitary-adrenal (HPA) axis function and brain plasticity (*Zucchi et al., 2014*). This assumption is supported by a recently published report

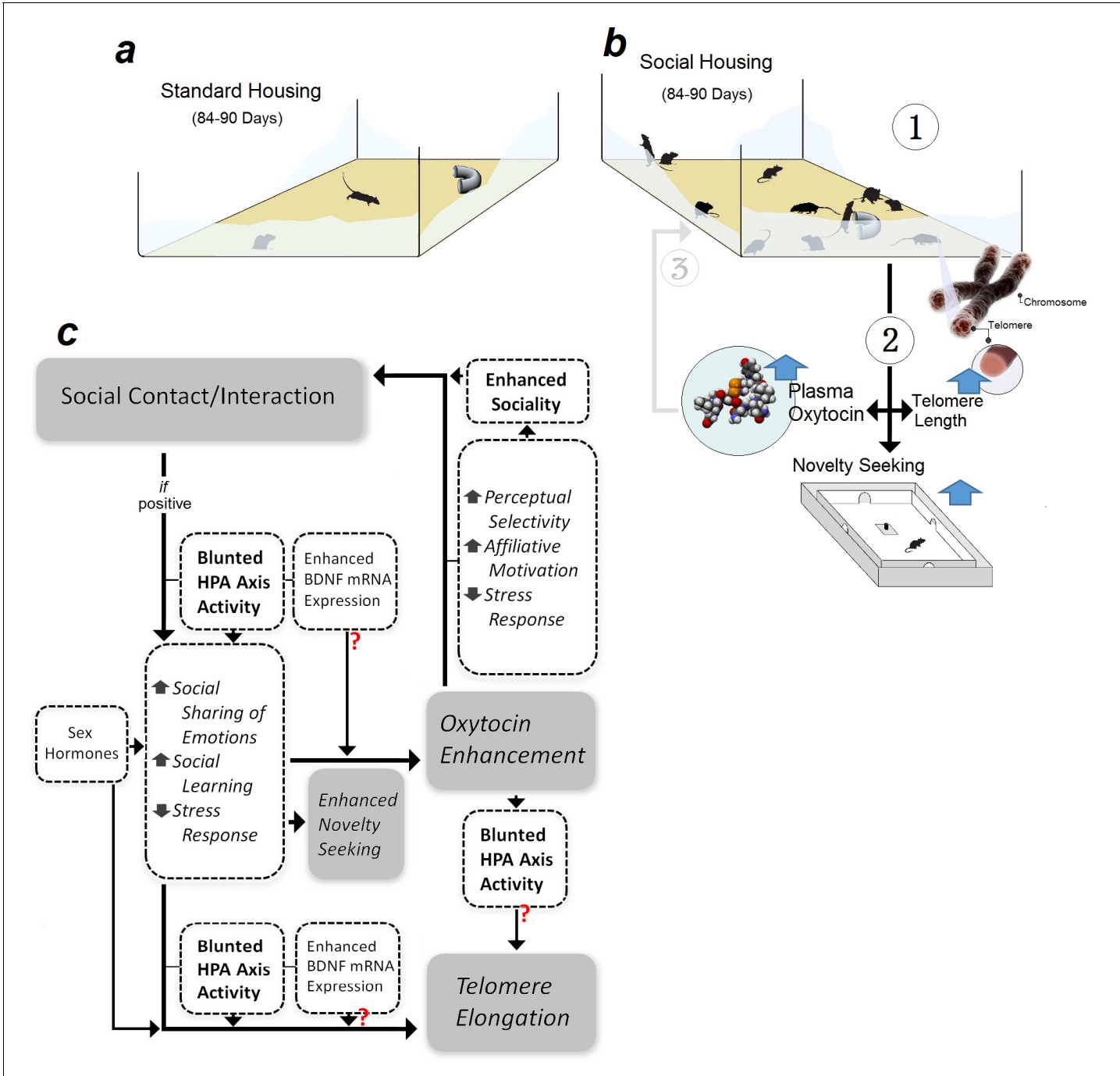

**Figure 5.** Representation of the experimental design and hypothetical mechanisms of social experience in rats. Male and female rats were raised in either (a) standard- or (b) social-housing units for 84–90 d. (b1) Prolonged social housing (b2) increased telomere length in females (TL) while enhancing plasma oxytocin (OT) in both sexes. Novelty-seeking behaviour in females more than males was responsive to social housing. (b3) Higher OT levels amplify social bonding and interaction through enhanced sociality. (c) Social interaction modulates novelty-seeking behaviours, OT, and TL along with HPA axis activity as a function of sex hormone status. Other hypothetical mechanisms to modulate social experience-dependent behaviour and neuroplasticity may include neurotrophic factors, such as brain-derived neurotrophic factor (BDNF).
DOI: https://doi.org/10.7554/eLife.40262.012

(*Zheng et al., 2014*) in which sensory experiences in mice promoted cortical development by up-regulation of synthesis and secretion of OT. Second, ongoing emotional sharing and social support foster prolonged social relationships (*Beery and Kaufer, 2015*) along with activation of the brain's dopamine reward system (*Rilling et al., 2002*). Both, regulating stress responses and rewarding social interaction, are vital corollaries in OT-mediated social engagement.

While OT is produced both centrally and peripherally, social cues generally cause OT releases centrally in the brain, which in turn modulates OT-receptor activities in the socially relevant circuitry. Sex-specific differences (*Joel et al., 2015*) in cortical regions including mPFC revealed higher OT receptor density and signaling in females (*Smith et al., 2017*; *Nakajima et al., 2014*). Accordingly, compared with males, vertebrate females appear sensitized to social support and interactions (*Beery and Kaufer, 2015*; *Ozbay et al., 2007*) in terms of emotional expressions (*Kring and Gordon, 1998*) and enhanced social cognition (*Gur et al., 2012*). The neurophysiological sequelae (including changes in cardiovascular, neuroendocrine, and immune function) (*Cohen and Wills, 1985*; *House et al., 1988*; *Lincoln et al., 2000*) via providing social supports, reduce the impact of stress responses to promote stress resiliency, especially in females (*Uchino, 2006*; *Charney, 2004*). Social behaviours and OT are linked to stress regulation through contextual (e.g., the duration of social life or presence of a familiar individual) and inter-individual factors (e.g., sex) (*Bartz et al., 2011*). The present changes after social experience for 3 mo address both types of intervening factors resulting in sex-specific regulation of social behaviour and exacerbated hormonal responses in female.

## Social experiences promote novelty-seeking behaviour

The present experiments provide evidence for the involvement of OT in exploratory and novelty-seeking behaviour (*Kazlauckas et al., 2005*) encouraged by the novel central object in the CFT. In agreement with our previous findings, female rats that experienced social enrichment across development showed extended exploration of the central areas in the CFT, thus increasing the radius of exploratory activity (*Faraji et al., 2014*). The latter indicates lowered stress response (*Faraji et al., 2014*), which agrees with the finding that socially raised female rats display lower HPA axis response and reduced anxiety-related behaviours (*Faraji et al., 2018*). Accordingly, intensified OT action through social enrichment enhances social bonding and reduces stress responses (*Swain et al., 2014*). Because OT may mediate sexual dimorphisms in regional neuroplasticity (*Hillerer et al., 2014*) and exert anxiolytic effects (*Ayers et al., 2011*), social experiences may affect brain anatomy and novelty-seeking behaviours especially in females.

OT-mediated enhancement in novelty-seeking behaviour may stem from the close relationship between OT receptors and central dopamine reward systems in mediating incentive response to novelty (*Bardo et al., 1996*). Although it has been suggested that females display generally lower levels of novelty seeking than males (*Hughes, 1968*), the present findings suggest that early life experiences and the endocrine consequences of social bonding may determine sex-dependent differences in novelty-seeking behaviours. Moreover, social relationships play key roles in the formation of emotional and social responsiveness that reduce aversive outcomes (e.g. fear and anger) in social interactions (*Ozbay et al., 2007*; *Averbeck, 2010*; *Sippel et al., 2015*). It appears that OT reciprocally influences the perceived social cues in a sex-dependent manner through these behavioural phenotypes (*Olff et al., 2013*).

## Social experiences increase TL in females via OT enhancement

Telomeres act to protect the ends of chromosomes from degradation, and consequently, control chromosomal stability and cellular senescence (*Mitchell et al., 2017*; *Drury et al., 2012*). Here, socially reared female rats, in addition to increased plasma OT concentration, had significantly longer TL than social males (~638.01 bp). By contrast, interruption of endogenous OT release induced TL attrition (~442.66 bp) in social females only. Despite females appearing to have longer TL than males (*Merrill et al., 2017*), TL elongation in social females offers a potential predictor for extended lifespan in humans (*Holt-Lunstad et al., 2010*) and rodents such as rats (*Yee et al., 2008*), who typically engage in social relationships. Interestingly, TL was recently shown to positively correlate with the number of surviving children born to a woman (*Fagan et al., 2017*), suggesting a possible parallel link between reproductive success and longevity in females.

The pathways through which telomere elongation is influenced by peripheral OT enhancement or inhibition are not yet fully understood. For the present results, axo-vasal contacts for endocrine neurosecretion (*Knobloch and Grinevich, 2014*), a prominent central-peripheral axis which releases OT, vasopressin (VP), and their homologues into systemic blood circulation, may explain how alterations in peripheral OT can influence TL measured in the samples collected from ear. Axonal release of OT in the brain and in the circulatory system seem to be coordinated as central OT and peripheral OT in animals can be released simultaneously (*Wotjak et al., 1998*). In humans, a correlation was shown between plasma and brain OT indicating that peripheral OT represents a suitable proxy of central OT (*Lefevre et al., 2017*). It is also important to consider that, as they share a common ontogenetic origin from ectoderm, both brain and skin tissues contain similar collagenous proteins, reticulin, vasculature, and connective tissues such as fibroblasts and adipocytes. Therefore, changes in peripheral TL in general, and skin cells in particular, predict TL alterations found in brain tissue (*Hehar and Mychasiuk, 2016*).

The present results revealed that exposure to the OT antagonist across development and during social experiences accelerates TL attrition only in females. A causal explanation of how social and neurohormonal processes interact to induce a sex-dependent effect on TL was beyond the scope of the experiments. Three lines of evidence, however, support the observation of TL alteration in socially enriched females: (1) correlation between OT and social behaviours in females is stronger than males (*Barraza and Zak, 2009*); (2) both social experiences and OT hormone blunt HPA axis activity (*Sippel et al., 2015*; *Ditzen et al., 2009*); and more importantly, (3) TL shortening is associated with dysregulation of HPA axis activity and abnormal levels of cortisol (*Schutte et al., 2016*). Hence, the pathways involved in social experience-induced OT responses may contribute to sex differences in TL in close interaction with the HPA system. Notably, social experiences in our previous findings (*Faraji et al., 2018*) reduced HPA axis activity in F0 female rats and the F1 non-social housing offspring born to social mothers, which suggests an intergenerational impact of social rearing history through regulation of the HPA response in female rats. Various other genetic mechanisms not investigated here may also determine social behaviours. Immediate-early genes, such as *zif268* mRNA expression in the dorsal and ventral medial mPFC in rats, appear to be sexually dimorphic (*Stack et al., 2010*). Because *zif268* expression can activate specific signaling pathways, it may lead to long-lasting synaptic changes in the brain. For example, rats show sexual dimorphism in neuronal firing rates in the barrel cortex in response to social interaction components (*Jurek and Neumann, 2018*).

Taken together, the present results suggest that a socially stimulating environment is especially critical to female health trajectories. Even short periods of OT inhibition make females vulnerable to TL shortening via OT-mediated dysregulation of HPA system activity. Although influenced by sex and gender (*Mather et al., 2011*), TL measurement may be a useful tool in future studies to understand and predict health and longevity particularly in females.

## Conclusion

Social experiences provide a source of psychoneurophysiological empowerment in both humans and animals and promote healthy development and successful aging (*Yang et al., 2016*). The present data causally associate social stimulation with genetic and behavioural improvements possibly as a function of endogenous OT. The findings indicate that early social experiences may last far into adulthood and manifest in TL elongation and novelty-seeking enhancement as a function of sex. The impact on cortical neuroplasticity, neurohormonal and behavioural outcome may be transmitted to the F1 non-social offspring in a sexually dimorphic manner. Thus, a socially stimulating environment may promote stress resiliency and mental health not only in exposed individuals, but also in their intergenerationally programmed descendants. Social enrichment may therefore provide a therapeutic avenue to promote stress resiliency and chances of healthy aging across generations.

## Materials and methods

### Animals

Male and female Wistar rats (222–430 g), bred and raised at the local vivarium were used in the present experiments. All animals were housed in a constant-temperature (21–24°C) room on a 12 h light/

dark cycle (lights on at 7:30 am) with ad libitum access to food and water. Rats were handled for approximately 3 min daily for 5 consecutive d prior to any experimental manipulations. Also, body weight in all animals was recorded every 4 d. The behavioural testing was performed during the light phase of the cycle, at the same time of day by three experimenters blind to the experimental groups. All procedures in this study were carried out in accordance with the National Institutes of Health Guide to the Care and Use of Laboratory Animals, and were approved by the institutional animal care committee (Protocol No. 004674BGH; Avicenna Institute of Neuroscience-AINS).

## Experimental design

Experiment 1: pups and their mothers were left undisturbed from postnatal day (PND) 1–21. After weaning at PND 21, 48 pups (five to six rats per litter) gathered from nine different litters were randomly assigned to four experimental groups in two housing conditions: (1) standard housing (males, n = 12), (2) standard housing (females, n = 12), (3) social housing (males, n = 12), and (4) social housing (females, n = 12). Novelty-seeking behaviour analysis included seven standard animals (four males, three females) and six social animals (three males, three females) from a previous experiment (*Barrett and Richardson, 2011*) to increase sample size. After living in either standard- or social-housing conditions for 86–89 d, all animals were subjected to blood sampling (days 83–86) and behavioural assessment (days 85–90). Animals were euthanized when behavioural assessments were completed. Experiment 2: at weaning, pups from 14 litters were randomly selected for Experiment 2. To accommodate OT antagonist (OT ANT) administration, animals were split into eight groups (n = 10–11/group): (1,2) standard housing: male and female (CONTROL), (3,4) standard housing: male and female (OT ANT), (5,6) social housing: male and female (CONTROL), and (7,8) social housing: male and female (OT ANT). All animals were housed in either standard- or social-housing units for 84–90 d and were subjected to blood sampling on days 80–83 and behavioural assessments on days 84–89. Rats were euthanized when behavioural testing was completed on days 88–91.

## Housing condition

A complete description of the housing condition was previously reported by this team (*Faraji et al., 2018*). Briefly, animals assigned to the social housing condition were housed and raised in groups of 10–11 within two separate social housing units (86 cm × 86 cm × 41 cm) with no additional environmental enrichment provided. Animals were constantly living in the same units until completion of the experiment. In contrast, rats assigned to standard housing conditions were housed and raised in non-sibling groups of two or three within Makrolon shoebox cages (86 cm × 86 cm × 41 cm). Animals were briefly removed from their environments when bedding material was changed. Animals in social and standard housing conditions were raised separated by sex.

## Blood sampling and oxytocin (OT) assay

Peripheral levels of OT (OTp) are assumed to reflect central releases of OT (OTc) in rats (*Wotjak et al., 1998*) and humans (*Lefevre et al., 2017*) with some procedural exceptions. Blood samples (0.5–0.7 mL) were taken 1–2 d prior to behavioural assessments (*Faraji et al., 2014*) to measure plasma concentration of OT using solid phase radioimmunoassay (RIA) (*Alburges et al., 2000*). Blood sample tubes containing aprotinin (500 kallikrein inactivation units/mL blood) were centrifuged at 3000 rpm (1700 g) for 15 min at 4°C. Plasma was stored at –70°C until analysis. Sample extraction and concentration were performed according to the manufacturer's manual provided with the kits (Phoenix Pharmaceuticals, Burlingame, CA), and a method previously described by Kobayashi et al. (*Kobayashi et al., 1999*). Intra- and inter-assay variability was 7% and 15%, respectively, as reported by the manufacturer. No behavioural testing was performed on blood sampling days.

## Oxytocin (OT) inhibition

The non-peptidyl OT antagonist (OT ANT) L-366,509 (MedKoo Biosciences, Inc., Morrisville, USA) (*Evans et al., 1992*; *Pettibone et al., 1993*) was administered (50 mg/kg) subcutaneously into the scruff of the neck in the OT ANT groups in Experiment 2. Administration occurred every other day (between 11:00 am and 12:00 noon) for 42–43 d (in total, 42–43 doses/rat) to intermittently inhibit or reduce OT secretion. Administration of L-366,509 started within the first week of the experiment (day 5) and ended 14 h before perfusion. The L-366,509 dosage used in Experiment 2 was chosen

based on the results in two pilot experiments (each n = 3–5) that were found to significantly reduce plasma OT levels, and a previous report (*Kobayashi et al., 1999*). Physiologic saline solution was injected subcutaneously in control groups (42–43 doses/rat) to avoid the confounding differential effect of stress resulting from repeated OT ANT injections.

Although there is no information available on the penetration of L-366,509 into the blood-brain barrier (BBB) in the literature, a new generation of spiroindenylpiperidine camphor-sulfinamide OT antagonist (i.e. L-368,899, (*Williams et al., 1994*)) which readily crosses the BBB (*Smith et al., 2010*) was developed from L-366,509 with the same structural and lipophilic molecular characteristics. Thus, it is likely that antagonist L-366,509 crossed the BBB.

## Corridor field task (CFT)

Novelty-seeking exploratory behaviour in the CFT was assessed according to the method of Faraji et al. (*Faraji et al., 2018*). Briefly, the floor of the CFT was divided into three zones: (1) corridor zone comprising the zone between external and internal walls; (2) open zone comprising the area within the task excluding corridor and the central zones; and (3) central zone comprising the middle area of the arena. Both variations of the CFT (plain CFT without a central object, and CFT with a central object) were used in the present study to assess locomotor aspects of free exploration. All rats were individually allowed to freely explore the environment for 8 min. Animals' performance was recorded under dim illumination by a ceiling-mounted camera (CCTV Auto tracking PTZ; SONY, Tokyo, Japan) and analyzed by a computer tracking system (SINA motiongraph, *Ayers et al., 2011*), Tabriz, Iran) through analysis of the time spent in each zone. The apparatus was cleaned with 70% alcohol between test sessions.

## Telomere measurement

Assuming that OT receptors are expressed in rat ear (*Kitano et al., 1997*) and human skin (*Deing et al., 2013*), genomic DNA from ear notch samples was extracted by a commercial kit (Sigma-Aldrich, Tokyo, Japan) based on the manufacturer's protocol (*Hehar and Mychasiuk, 2016*). Telomere length (TL) was measured using a modified protocol for a quantitative Real-Time Polymerase Chain Reaction (qRT-PCR) assay based on previous reports (*Hehar and Mychasiuk, 2016*; *Cawthon, 2002*). The protocol enabled calculation of the ratio of telomere copy repeats to a single-copy reference gene (36B4) to determine relative TL. Accordingly, when the ratio was found equal to 1.00, the unknown DNA was assumed to be identical to the reference DNA, whereas the ratios greater to, or less than one, were - respectively - considered increased or decreased telomere repeat numbers. A pipetting robot was used for all assay runs to avoid pipetting errors and inconsistency in reactions, and PCR assays were performed by two independent technicians. The relative TL was determined based on the linear regression equation reported by Cawthon (*Cawthon, 2002*).

## Statistical analysis

Effects of main factors (housing condition; two levels, group; four and eight levels, sex; two levels, Zone; three levels, litter; nine and 11 levels) were analyzed separately for the time spent in CFT, with and without central object by repeated-measure and one-way ANOVA, and multivariate analysis of variance (MANOVA). Also, post-hoc test (Tukey) was used to adjust for multiple comparisons. Familywise error was considered prior to the multiple post-hoc analyses if necessary. Data for OT levels and TL were also analyzed by separate ANOVA analyses with the main between-subject factors of group and sex. Linear regression analysis was performed in Experiment 1 to predict the outcome (dependent) variables (the novelty-seeking behaviour in the CFT and telomere length) through the predictor (independent) variable (OT levels). Dependent and independent sample t-tests were conducted when necessary. In all statistical analyses (SPSS 16.0, SPSS Inc., USA), a p-value of less than 0.05 was considered statistically significant. Values represent mean ±SEM.

## Acknowledgements

The authors are grateful to Tanzi Hoover for productive comments on the manuscript. We thank B Rezazadeh, J Foroughi, and F Nejad-Ghorban for their assistance in behavioural testing, Dr. GR Namazi for technical assistance, and colleagues at the research ethics board at the Avicenna Institute of Neuroscience (AINS) for suggestions and comments. The authors would also like to thank the

animal care staff at the AINS vivarium for assistance with animal husbandry. Special thanks to Dr. M Fakhamati for his suggestions and assistance with statistical analysis. Funding for this study was provided by a basic science research program-AINS (#41108–010) to RM, and by Natural Sciences and Engineering Research Council of Canada Discovery Grant #5519 to GM.

## Additional information

### Funding

| Funder | Grant reference number | Author |
|---|---|---|
| Natural Sciences and Engineering Research Council of Canada | 5519 | Gerlinde AS Metz |
| Avicenna Institute of Neuroscience | Basic Research Program 41108–010 | Reza Moeeini |

The funders had no role in study design, data collection and interpretation, or the decision to submit the work for publication.

### Author contributions

Jamshid Faraji, Conceptualization, Methodology; Mitra Karimi, Supervision, Investigation, Visualization, Project administration; Nabiollah Soltanpour, Conceptualization, Data curation, Supervision, Investigation, Methodology, Project administration; Alireza Moharrerie, Formal analysis, Supervision, Investigation, Project administration; Zahra Rouhzadeh, S Abedin Hosseini, S Yaghoob Jafari, Investigation, Project administration; Hamid lotfi, Supervision, Investigation, Project administration; Shabnam Roudaki, Formal analysis, Supervision, Investigation, Methodology, Project administration; Reza Moeeini, Conceptualization, Resources, Data curation, Formal analysis, Supervision, Validation, Methodology, Project administration, Writing—review and editing; Gerlinde AS Metz, Conceptualization, Resources, Supervision, Methodology, Writing—review and editing

### Author ORCIDs

Jamshid Faraji (iD) https://orcid.org/0000-0002-1726-5836
Gerlinde AS Metz (iD) http://orcid.org/0000-0002-5801-7716

### Ethics

Animal experimentation: All procedures in this study were carried out in accordance with the National Institute of Health Guide to the Care and Use of Laboratory Animals, and were approved by the institutional animal care committee (Protocol No. 004674BGH; Avicenna Institute of Neuroscience-AINS).

### Decision letter and Author response

Decision letter https://doi.org/10.7554/eLife.40262.015
Author response https://doi.org/10.7554/eLife.40262.016

## Additional files

### Supplementary files

• Transparent reporting form
DOI: https://doi.org/10.7554/eLife.40262.013

### Data availability

All data generated or analysed during this study are included in the manuscript and supporting files. Source data files have been provided for all figures.

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
