## [Decision Letter]

[Editors’ note: this article was originally rejected after discussions between the reviewers, but the authors were invited to resubmit after an appeal against the decision.]

Thank you for submitting your work entitled "Oxytocin-mediated Social Enrichment Promotes Longevity and Novelty Seeking" for consideration by *eLife*. Your article has been reviewed by four peer reviewers, including Peggy Mason as the Reviewing Editor and Reviewer #1, and the evaluation has been overseen by a Senior Editor. The following individuals involved in review of your submission have agreed to reveal their identity: Takefumi Kikusui (Reviewer #3).

Our decision has been reached after consultation between the reviewers. Based on these discussions and the individual reviews below, we regret to inform you that your work will not be considered further for publication in *eLife*.

The reviewers' chief concern is the lack of an OXT antag control. It is plausible that the stress of the non-trivial number of injections was the active agent here rather than the pharmaceutical compound. If the authors came back with such a control, we would be interested in such a (new) submission. The authors may also benefit from a number of other comments from the reviewers including walking back some of the conclusions to acknowledge the correlational nature of the data as well as the lack of a compelling direct path from housing to telomere length regulation.

Reviewer #1:

There is a lot to like and a fair amount to easily worry about in this study. The big positives are the large groups compared to normal social housing. Comparing normal (yet deprived in the natural scheme of things) to enriched (probably close to true normal for rats) is a strong design, far better than comparing 2-4 rodent groups (deprived) to social isolation (a ridiculous condition with no ecological correlate). The investigation of brave behavior (going in to the OF center with or without an object there) across the sexes and housing conditions is an interesting design aimed at important questions regarding early social development and sexual dimorphisms. The biggest issue with this paper is the lack of an injection control for the OXT antag injections. The results obtained – less exploration of the center after the antag – could easily result from a differential effect of stress (the stress of 28 injections!!) on females compared to males, Is this issue surmountable with the data in hand? I fear no but look forward to seeing what the other reviewers think.

At the outset of the results it is necessary to state how many rats are in social (n=10-11) and standard (n=2-3) groups. It would also be useful to state how the rats from different litters were assigned to the groups. The no difference reported across litter is not interpretable without this information.

Then the authors need to be far more careful about the use of the word "group." They state "…across different groups (n=21-23/group)" which appears to say that there are 21-23 in the group housed cages but in point of fact this appears to be the number of animals in the social or standard conditions. Which would imply that two Social groups (n=20-22 rats), one for each sex, were compared to 8-10 Standard groups, half for each sex? This should be explicitly stated. Then the number of rats in Figure 1A Social as 44 and in 1B as 54. Please make sense of these various numbers.

A significant effect of Housing Condition (F1,86=24.85, p{less than or equal to}0.001) and Group (F3,84=13.69, p{less than or equal to}0.001). By Group do the authors mean sex? I thought the conditions were the groups. This is very confusingly written. I think the authors are saying there are 4 groups whereas it should be a 2x2 factor design with housing – 2 conditions – and sex – 2 groups.

It appears that the groups are mixed males and females. Are some of the females then pregnant? If this is the case, then some of the females must be pregnant which would obviously change their behavior. Please explain.

There is no saline control for the OT antag injections. This is an experimental choice that must be justified and is possibly a deal-killer. The injections were "every other day (between 11:00-12:00 am) for 13 days (in total, 28 doses/rat)…" Clearly every other day for 13 days does not yield 28 doses per rat. So, this needs to be rewritten into careful English so that the reader is told what was done.

Reviewer #2:

This is a terrific paper that should be published as soon as possible. I believe it stands to make a substantive contribution to the field and will be heavily cited. It explains the interactions among social relationships, Oxytocin, telomeres and longevity convincingly. Links to HPA axis function are also very interesting. The implications for human relationships are very intriguing. The background and discussion are soundly presented. The analytic methods are sound, and the results are compelling.

I am less able to review the data collection methods, as I am not an animal scientist, so I trust other reviewers to do so.

There were no grammatical or typos to note.

Again, I recommend publication as soon as possible.

Reviewer #3:

This paper demonstrated that long-term social housing in the juvenile period enhanced peripheral OT secretion, together with longer TL. Novelty-seeking behavior was also increased by social housing. These housing effects were more robust in females. The data are interesting because the authors examined causality between OT secretion, TL and behavior. Even though there is no neuron/ brain region analysis conducted, the findings would reveal a new function of OT.

Here are my concerns,

1) Ear notch samples were used to assess TL, but is there any evidence showing that these cells express oxytocin receptor? If so, please add the reference in the Materials and methods section. If not, the authors need to include the discussion how blood OT can affect TL in these OTR-lacking cells.

2) OT antagonist L-366,509 was used to inhibit OT system in the CNS. Please add the information about penetration of this chemical through blood brain barrier in the Materials and methods section.

3) The authors mentioned the function of OT on the HPA axis, but this paper did not include the data of corticosterone. If the blood samples remain, please measure corticosterone, to make the conclusion more convincing.

4) Related to #2 and #3, higher corticosterone exposure will shorten the TL in peripheral cells. Please add the corticosterone data if possible.

5) Were the changes observed in social-housing rats caused by social interactions? If the rats were housed in a large area, the locomotor activity can increase. In this experimental setting, the authors cannot conclude that the changes were caused by social interactions. There might be an appropriate control group needed, such as similar locomotor activity without social interaction. If the authors cannot exclude the effect of activity (physical exercise can facilitate OT secretion, TL etc.), the discussion should be modified.

6) The social housing was conducted just after the weaning, so the effects of social housing would be robust (still in the developmental period). If the social housing were to be conducted in the adulthood or aged rats, could the same positive affects be observed? In humans, social loneliness in elderly is one of the important social issues related to the health problems.

Reviewer #4:

In manuscript by Faraji et al., the authors reported correlations between circulating oxytocin concentration and prolonged social housing, in both male and female rats. They further showed that prolonged social housing correlated with elongated telomere length in only in female rats, an effect blocked by the oxytocin antagonist L-366,509. While the correlation between oxytocin and telomere length is potentially interesting, I believe that there are some gaps of logic in the author's interpretation of the results.

1) The title states that "oxytocin-mediated social enrichment promotes longevity and novelty seeking". The authors did not measure life span of their experimental animals, only telomere length. To my understanding, telomere length is not equivalent to longevity and thus I find the title misleading.

2) This is my main concern: the authors showed that social housing increased oxytocin (OT) level in both male and female rats, but only increased telomere length in female rats. Administration of the OT antagonist blocked the effect of social housing in both males and females, but only blocked TL increase in social females. Based on these results, the authors concluded that "…extended social experiences modulate TL and novelty seeking through the OT system in a sex-dependent manner…" (Abstract). These is extensive literature reporting oxytocin as a "social hormone", thus the correlation between oxytocin level and social housing is consistent with other published results. To my knowledge, the link between oxytocin level and telomere length has not been previously reported and certainly there are no proposed mechanisms. Under these circumstances, I would be more willing to accept the authors' correlative results as potentially interesting and significant, if the correlations between oxytocin level and telomere length were similar to those of social housing, i.e. either both correlations held for males and females, or both held for only females.

3) I do not know the telomere field, but is telomere length in the ear representative of the entire body? Would telomere length in multiple cell types need to be measured to reach a conclusion?

4) I do not agree with the authors' interpretation of their results, in statements such as: "The present study demonstrates a key role of OT in socially mediated behavioral fitness and telomere length as an indicator of overall health and longevity" (Discussion section); "…Thus OT is causally involved in mediating beneficial effects of social experience on behavior and TL…" (Discussion section), and "The present data causally associates social stimulation with genetic and behavioral improvements as a function of endogenous OT" (subsection “Conclusion”). The authors did not measure behavior fitness, health or longevity. Based on arguments presented in point 2 above, I don't think the results provided sufficient evidence to demonstrate a causal link between OT and TL. In order to demonstrate a causal link, at a minimum, the authors would need to demonstrate that in vivo OT administration can promote telomere length, and that the effect of social housing on telomere length are abolished in oxytocin knockout mice. Telomere length needs to be measuring in multiple tissues and not just ear punches. Also, no statements can be made regarding behavior fitness, health or longevity unless data is provided.

5) Related to points 2 and 4 and given the discussion regarding how accurately oxytocin level in the plasma measured using RIA or ELISA reflects oxytocin level in the brain, the authors could measure oxytocin mRNA level or immunostain for oxytocin in brain sections, to further establish the correlation between oxytocin level and telomere length.

6) Results presented in Figure 2A-D are very similar to that presented in a previous publication from the same lab (Faraji et al., 2018). The authors should reference their previous results in the Results section, and not just the Materials and methods section.

7) Statements such as "…Thus, social females' exploration in the corridor zone decreased -1.84 for each pM of OT concentration" (subsection “Novelty-Seeking Behaviour in Females was Significantly Influenced by Social Experiences”), and "…TL, therefore, only in social females increased 8.53 bp for each pM of OT level" (subsection “Novelty-Seeking Behaviour in Females was Significantly Influenced by Social Experiences”), do not make sense to me, as the data points for TL range from 4000 to 5700 bp, and that for OT range from around 40 to 125 pM. In light of huge differences in the levels of OT and TL, these precise interpretations do not make much sense to me.

[Editors’ note: what now follows is the decision letter after the authors submitted for further consideration.]

It is great that you did saline controls. Of course, it is a problem that none of the reviewers realized that. Please address this by making it far more evident in the manuscript. Also make sure that the saline groups make their way into the figures. Finally, please respond to the other issues raised by the reviewers. We look forward to your resubmission.

---

## [Author Response]

[Editors’ note: the author responses to the first round of peer review follow.]

Our decision has been reached after consultation between the reviewers. Based on these discussions and the individual reviews below, we regret to inform you that your work will not be considered further for publication in eLife.The reviewers' chief concern is the lack of an OXT antag control. It is plausible that the stress of the non-trivial number of injections was the active agent here rather than the pharmaceutical compound. If the authors came back with such a control, we would be interested in such a (new) submission. The authors may also benefit from a number of other comments from the reviewers including walking back some of the conclusions to acknowledge the correlational nature of the data as well as the lack of a compelling direct path from housing to telomere length regulation.Reviewer #1:There is a lot to like and a fair amount to easily worry about in this study. The big positives are the large groups compared to normal social housing. Comparing normal (yet deprived in the natural scheme of things) to enriched (probably close to true normal for rats) is a strong design, far better than comparing 2-4 rodent groups (deprived) to social isolation (a ridiculous condition with no ecological correlate). The investigation of brave behavior (going in to the OF center with or without an object there) across the sexes and housing conditions is an interesting design aimed at important questions regarding early social development and sexual dimorphisms. The biggest issue with this paper is the lack of an injection control for the OT antag injections. The results obtained – less exploration of the center after the antag – could easily result from a differential effect of stress (the stress of 28 injections!!) on females compared to males, Is this issue surmountable with the data in hand? I fear no but look forward to seeing what the other reviewers think.

As stated in the manuscript (Materials and methods section), physiologic saline solution was injected in control groups. The new paragraph now reads:

“…Physiologic saline solution was injected subcutaneously in control groups (42-43 doses/rat) to avoid the confounding differential effect of stress resulted by repeated OT ANT injections.”

We also included the word “saline” in control group terminology in Figures 3 and 4 in order to highlight the saline injections in all control animals in Experiment 2 (please see Figure 3A-D, and Figure 4A-D, F and H).

At the outset of the results it is necessary to state how many rats are in social (n=10-11) and standard (n=2-3) groups. It would also be useful to state how the rats from different litters were assigned to the groups. The no difference reported across litter is not interpretable without this information.

We appreciate this comment and haveaddressed this concern by re-writing the first sentence of the Results section in the manuscript as the reviewer suggested. The new sentence now reads:

“…Figure 1A shows the changes in plasma OT concentrations across different groups as a function of housing condition (Standard, *males*: n=23, *females*: n=21; Social, *males*: n=22, *females*: n=22)…”.

Also:

“…Examination of TL in ear notch skin cells showed that socially-raised rats (*males*: n=31, *females*: n=23) had greater telomere length than standard animals (*males*: n=28, *females*: n=27;…” (Results section).

Moreover, to further address the reviewer’s concerns we have re-written the sampling method for different litters in the manuscript that now reads:

“…Pups and their mothers were left undisturbed from postnatal day (PND) 1–21. After weaning at PND 21, 48 pups (5-6 rats per litter) gathered from 9 different litters were randomly assigned to four experimental groups in two housing conditions:…”(Materials and methods section).

Then the authors need to be far more careful about the use of the word "group." They state "…across different groups (n=21-23/group)" which appears to say that there are 21-23 in the group housed cages but in point of fact this appears to be the number of animals in the social or standard conditions. Which would imply that two Social groups (n=20-22 rats), one for each sex, were compared to 8-10 Standard groups, half for each sex? This should be explicitly stated. Then the number of rats in Figure 1A Social as 44 and in 1B as 54. Please make sense of these various numbers.

We have re-written this part of the Results section to address the reviewer’s concerns that now reads:

“Figure 1A shows the changes in plasma OT concentrations across different groups as a function of housing condition (Standard, *males*: n=23, *females*: n=21; Social, *males*: n=22, *females*: n=22). A significant effect of Housing Condition was observed in terms of circulating OT concentration (Social vs. Standard; F_1,86_=24.85, p≤0.001) and Group (social males and females vs. standard males and females; F_3,84_=13.69, p≤0.001), suggesting…”..

A significant effect of Housing Condition (F1,86=24.85, p{less than or equal to}0.001) and Group (F3,84=13.69, p{less than or equal to}0.001). By Group do the authors mean sex? I thought the conditions were the groups. This is very confusingly written. I think the authors are saying there are 4 groups whereas it should be a 2x2 factor design with housing – 2 conditions – and sex – 2 groups.

Please, refer to the previous comments. Also, we have included further information into the Statistical Analysis about how we defined and analyzed variables in order to address this major concern. The new inclusion now reads:

“…Effects of main factors (*Housing Condition*; 2 levels, *Group*; 4 & 8 levels, *Sex*; 2 levels, *Zone*; 3 levels, *Litter*; 9 & 11 levels) were analyzed separately for the time spent in CFT, with and without …” (Materials and methods section).

It appears that the groups are mixed males and females. Are some of the females then pregnant? If this is the case, then some of the females must be pregnant which would obviously change their behavior. Please explain.

The authors apologize for not making this clear enough in the manuscript. Males and females were raised separately either in social or standard housing conditions. This is mentioned in the revised manuscript in the Materials and methods section for further explanation.

There is no saline control for the OT antag injections. This is an experimental choice that must be justified and is possibly a deal-killer.

We have addressed these concerns by including a new paragraph. The new paragraph now reads:

“…Physiologic saline solution was injected subcutaneously in control groups (42-43 doses/rat) to avoid the confounding differential effect of stress resulted by repeated OT ANT injections.” (Materials and methods section).

We also included the word “saline” in control group terminology in Figures 3 and 4 in order to highlight the saline injections in all control animals in Experiment 2 (please see Figure 3A-D, and Figure 4A-D, F and H).

The injections were "every other day (between 11:00-12:00 am) for 13 days (in total, 28 doses/rat)…" Clearly every other day for 13 days does not yield 28 doses per rat. So, this needs to be rewritten into careful English so that the reader is told what was done.

Many thanks for the comment, and we apologize for this mistake. We rephrased the sentence that now reads:

“…was administered (50 mg/kg) subcutaneously into the scruff of the neck in the OT ANT groups in Experiment 2. Administration occurred every other day (between 11:00-12:00 am) for 42-43 days (in total, 42-43 doses/rat) to intermittently inhibit or reduce OT secretion.…” (Materials and methods section).

Reviewer #3:This paper demonstrated that long-term social housing in the juvenile period enhanced peripheral OT secretion, together with longer TL. Novelty-seeking behavior was also increased by social housing. These housing effects were more robust in females. The data are interesting because the authors examined causality between OT secretion, TL and behavior. Even though there is no neuron/ brain region analysis conducted, the findings would reveal a new function of OT.Here are my concerns,1) Ear notch samples were used to assess TL, but is there any evidence showing that these cells express oxytocin receptor? If so, please add the reference in the Materials and methods section. If not, the authors need to include the discussion how blood OT can affect TL in these OTR-lacking cells.

This is a wonderful suggestion which we take very seriously. Indeed, we found out that at least in three previously published papers by Kitano et al., (1997), Bolan and Goren, (1987) and Deing et al., (2013), OT receptors were found to be expressed in the rat inner ear and epididymal adipocytes, and human skin. Two of them are cited now in the manuscript (please see Materials and methods section) to address the reviewer’s concerns. Moreover, the authors hypothesize that one possible mechanism in bridging increased plasma OT concentration to telomere elongation in the present experiments likely relies upon axo-vasal contacts for endocrine neurosecretion (please see Knobloch and Grinevich, 2014). By its traditional definition, endocrine neurosecretion refers to release and delivery of OT, VP and their homologues to systemic blood circulation. It is traditionally assumed that blood stream carries these molecules (e.g. OT) to their receptors which are abundantly expressed in peripheral organs such as uterus, mammary glands, intestinal tracts, heart, and more specifically in this case, even in the skin. Thus, we anticipate that a major part of the observed changes in TL induced by prolonged social experiences and the following OT enhancement is influenced by the axo-vasal contacts, which reveals an alternative explanation of the dynamic dialogue between the central capacities and peripheral repertoires in response to social experiences.

Also, to further address the reviewer’s concerns, we included a new paragraph in the discussion that now reads:

“…The pathways through which telomere elongation is influenced by peripheral OT enhancement or inhibition are not yet fully understood. For the present results, axo-vasal contacts for endocrine neurosecretion^14^, a prominent central-peripheral axis which releases OT, vasopressin (VP) and their homologues into systemic blood circulation, may explain how alterations in peripheral OT can influence TL measured in the samples collected from ear…” (please see subsection “Social Experiences Increase TL in Females via OT Enhancement**”**).

2) OT antagonist L-366,509 was used to inhibit OT system in the CNS. Please add the information about penetration of this chemical through blood brain barrier in the Materials and methods section.

We appreciate this thoughtful comment. To our best knowledge, there is no detailed proof available in the literature for the penetration of antagonist L-366,509 through the blood-brain barrier. However, based on experimental evidence, we assumed that L-366,509 crosses the blood-brain barrier. First, all lipophilic molecules readily cross the blood-brain barrier, even though hydrophilic molecules including OT itself do not (please see McEwen, 2004; Churchland and Winkielman, 2012). Because camphor-based non-peptide OT antagonists such as L-366,509 belong to bulky, lipophilic groups (see Evans et al., 1992) that have a high tolerance and considerably improved solubility in aqueous media, we anticipate that antagonist L-366,509 crosses the blood-brain barrier. Second and more importantly, the non-peptide OT antagonist L-368,899 is a newer member of structural classes of spiroindenes that is structurally derived from L-366,509 (see Williams et al., 1994). The most prominent difference between these two antagonists, however, refers to the higher affinity of L-368,899 rather than L-366,509 for OT receptors. Hence, as L-368,899 is reported to cross the blood-brain barrier with selective accumulation in specific areas of the limbic system where it might inhibit OT-induced social behaviours (see Smith et al., 2010), it allowed the authors to predict that antagonist L-366,509 can also cross the blood-brain barrier.

We fully agree that this issue has to be reported in the manuscript, and we apologize for this shortcoming. Therefore, we addressed the reviewer’s concern in the manuscript in a new paragraph that reads:

“…Although there is no information available on the penetration of L-366,509 into the blood-brain barrier (BBB) in the literature, a new generation of spiroindenylpiperidine camphor-sulfinamide OT antagonist (Williams et al., 1994) which readily crosses the BBB (Smith et al., 2010) was developed from L-366,509 with the same structural and lipophilic molecular characteristics. Thus, it is likely that antagonist L-366,509 crossed the BBB.” (please see Materials and methods section).

3) The authors mentioned the function of OT on the HPA axis, but this paper did not include the data of corticosterone. If the blood samples remain, please measure corticosterone, to make the conclusion more convincing.

The reviewer is absolutely right that CORT data could potentially provide support for the current results. Unfortunately, due to economic instabilities in Iran and other practical constraints we were unable to do these analyses. We deeply regret this issue as we indeed had initially planned on these analyses. Instead, to compensate for the lack of information of CORT changes, we now discuss our results in the light of previous relevant work by others and by our team in the Discussion section. The considerable amount of existing evidence provides solid support for our current findings. Again, we very much apologize for this shortcoming.

4) Related to #2 and #3, higher corticosterone exposure will shorten the TL in peripheral cells. Please add the corticosterone data if possible.

Please, see comment #3.

5) Were the changes observed in social-housing rats caused by social interactions?

This is an excellent point for consideration, thank you. We agree with the reviewer that both experiments in the present study could be conducted with more controls to better control for potentially confounding variables, even though we have done our best to have most of the confounding variables controlled. Also, it is very difficult to mention that changes observed in the present experiments were only the results of social interactions, because other uncontrolled external factors simultaneously may confound and/or overshadow our results. However, we strongly believe that social interactions as the most prominent component of social-housing condition designed here played a key role in the changes reported in the present study. Please refer to our logics in the next part of our answer to this comment.

If the rats were housed in a large area, the locomotor activity can increase. In this experimental setting, the authors cannot conclude that the changes were caused by social interactions. There might be an appropriate control group needed, such as similar locomotor activity without social interaction. If the authors cannot exclude the effect of activity (physical exercise can facilitate OT secretion, TL etc.), the discussion should be modified.

As stated in the manuscript (please see Materials and methods section), the same size of cage (86 cm × 86 cm × 41 cm) was used for the social and standard housing conditions, thus varying only the number of cage mates (standard, n = 2–3 vs. social, n =10- 11) while keeping the size of the environments constant (please see Faraji et al., 2018 for further details on the housing conditions). For two reasons, therefore, we do not agree with the reviewer that the observed changes are influenced by the enhanced level of physical activities in social-housing condition. First, if physical activity could determine the incidence or the extent of changes, all of these changes should be normally observed in the standard animals (here, controls) who were raised in the large but less crowded units. Second, changes in OT levels in response to physical activity induced by voluntary wheel running in rats are not entirely evident, yet (for instance, please see Broderick et al., 2014 and Bakos et al., 2007). Moreover, physical activity, to our knowledge, has been shown to impact OT levels in humans only if associated with a more strenuous type of physical exercise (see also Frisen et al., 2017 and Landgraf et al., 1982, for example). No additional enrichment stimulus such as a running wheel was provided for rats in neither standard nor social housing conditions. Further follow-up studies that address these issues would make a timely contribution to the literature.

6) The social housing was conducted just after the weaning, so the effects of social housing would be robust (still in the developmental period). If the social housing were to be conducted in the adulthood or aged rats, could the same positive affects be observed? In humans, social loneliness in elderly is one of the important social issues related to the health problems.

This is a very interesting question and worthy a follow-up study. Unfortunately, the present data do not allow a conclusion on social enrichment effects in older age, although previous data suggest that this assumption is very reasonable and worthy of further consideration (For example, please see Okabayashi’s report about social interactions and their effects on subjective well-being among Japanese elders; Okabayashi and Houghman, 2014).

Reviewer #4:In manuscript by Faraji et al., the authors reported correlations between circulating oxytocin concentration and prolonged social housing, in both male and female rats. They further showed that prolonged social housing correlated with elongated telomere length in only in female rats, an effect blocked by the oxytocin antagonist L-366,509. While the correlation between oxytocin and telomere length is potentially interesting, I believe that there are some gaps of logic in the author's interpretation of the results.1) The title states that "oxytocin-mediated social enrichment promotes longevity and novelty seeking". The authors did not measure life span of their experimental animals, only telomere length. To my understanding, telomere length is not equivalent to longevity and thus I find the title misleading.

We appreciate this comment and changed the title. The revised title now reads:

Oxytocin-mediated social enrichment promotes longer telomeres and novelty seeking

2) This is my main concern: the authors showed that social housing increased oxytocin (OT) level in both male and female rats, but only increased telomere length in female rats. Administration of the OT antagonist blocked the effect of social housing in both males and females, but only blocked TL increase in social females. Based on these results, the authors concluded that "…extended social experiences modulate TL and novelty seeking through the OT system in a sex-dependent manner…" (Abstract). These is extensive literature reporting oxytocin as a "social hormone", thus the correlation between oxytocin level and social housing is consistent with other published results. To my knowledge, the link between oxytocin level and telomere length has not been previously reported and certainly there are no proposed mechanisms. Under these circumstances, I would be more willing to accept the authors' correlative results as potentially interesting and significant, if the correlations between oxytocin level and telomere length were similar to those of social housing, i.e. either both correlations held for males and females, or both held for only females.

We would like to address this thoughtful and challenging comment by discussing several main arguments. First and most importantly, we believe that the experimental design of the present study enables us to make causal conclusions for the most parts of the results. Therefore, although for some experimental constraints we did not identify and/or offer a causal mechanism to connect variations in dependent variables such as peripheral OT levels to the TL, our data are still able to profile a clear “causal effect” for the independent and dependent variables, beyond the mere correlational conclusion. Indeed, the presence of control group(s) in both experiments assures empirical correlation along with temporal priority of the independent variable, thus providing a solid causal effect. Like the reviewer, the authors are also concerned about spurious data interpretation. Moreover, heavily standardized conditions by which we raised the groups of animals in standard units and aged-matched groups in a social environment helped us to interpret most variations in dependent variables (behavioural, hormonal and genetic) in the light of a causal conclusion. We believe that if we could (1) provide evidence of empirical association between at least one independent variable and one dependent variable, (2) confirm temporal priority of the independent variable (here, housing condition) using control groups, and also (3) tried to have nonspurious changes which were not influenced by irrelevant variable(s), the minimum criteria for making an experimental casual effect are met. Therefore, we believe that as long as we are able to manipulate the independent variable in a systematic manner and address internal and external control variables using careful implementation and analysis, the causal conclusion (*if* A [where A is social experience], *then* B [where B is OT enhancement and/or telomere elongation) can be possible and suggested.

Second, we agree with the reviewer that “…these is extensive literature reporting oxytocin as a "social hormone", thus the correlation between oxytocin level and social housing is consistent with other published results…”. However, to our knowledge, most of these correlational reports are depicting a hormone-to-behaviour approach (as stated in the manuscript) through which subjects are exposed to OT first, and then they are monitored for possible behavioural impacts of OT. In the present experiment, by contrast, we have chosen an alternative, behaviour-to-hormone approach by which we examined the impact of prolonged social experience on OT levels. This approach provides a new argument of causality to our results and conclusions as opposed to most literature about the experimental administration of OT and its consequences.

Third, we regard our data as considerably valid to causally link social lifestyle in early development (the major independent variable) among females to behavioural, endocrine and genetic outcomes (dependent variables). This seems more important when such changes are questioned in response to interruptions in social relationships among females and their health consequences, or when factors that in general determine telomere length in early life are investigated. Also, simultaneous responses (enhanced novelty seeking, increased levels of OT and telomere elongation) of females to persistent social experiences have implications in future investigations to causally hypothesize these changes. Examples include sex hormones that were previously shown to play a key role in telomere length (see Calado et al., 2006, for example) or for neurotrophins (e.g. BDNF) that are confirmed to influence telomere function (see Niu and Yip, 2011) and OT receptors in women (see Chagnon et al., 2015).

3) I do not know the telomere field, but is telomere length in the ear representative of the entire body? Would telomere length in multiple cell types need to be measured to reach a conclusion?

These are very exciting questions. The answer to both questions is “not necessarily”, even though the TL in lifespan is gender-related and follows a tissue-specific regulation during development. Indeed, the TL within individual tissues is regulated independently. In humans, TL seems highly variable between tissues and individuals. However, samples taken from peripheral organs such as kidney, liver, lung, etc. for inter-tissue comparison in intact rats are showing the same profile of changes in the TL (please, see Cherif et al.,’s work, such as Cherif et al., 2003). Also, studies with various strains of mice did not detect any differences in TL between tissues such as liver, testes or kidney (for instance, please see Coviello-McLaughlin and Prowse, 1997), although the TL generally differs between species. Moreover, TL in hearts and lungs in rats are equally affected by physical stress (please, see Wang et al., 2018) meaning that peripheral organs are possibly showing similar telomeric responses to experimental manipulations. Therefore, because different tissue types may represent different deterioration processes along with variable mortality, inter-tissue analysis in humans can potentially create more valid assessment of TL alterations in response to developmental influences or environmental manipulations. However, this does not seem to be necessarily required in rats as mentioned above. The only exception is for the TL in the brain tissues in rats and mice that were shown to depict different rates of TL alterations when compared with, for instance, spleen or other peripheral organs. Importantly, the reasons by which we have chosen the ear notch samples in the present study, in addition to the ease of collection, was that samples from ear are emerged from ectoderm, the same lineage as brain tissue. Like brain tissue, skin samples also contain a mixture of collagenous proteins, reticulin, vasculature, and connective tissues such as fibroblasts and adipocytes. Therefore, changes in rat peripheral TL in general, and skin cells in particular not only may represent TL changes in other organs influenced by experimental interventions, but also arguably predict TL alterations in the brain tissue (please see Hehar and Mychasiuk, 2016). To address this concern, we have included a new statement in the Discussion section that now reads:

“…It is also important to consider that, as they share a common ontogenetic origin from ectoderm, both brain and skin tissues contain similar collagenous proteins, reticulin, vasculature, and connective tissues such as fibroblasts and adipocytes. Therefore, changes in peripheral TL in general, and skin cells in particular, predict TL alterations found in brain tissue (Hehar and Mychasiuk, 2016).

4) I do not agree with the authors' interpretation of their results, in statements such as: "The present study demonstrates a key role of OT in socially mediated behavioral fitness and telomere length as an indicator of overall health and longevity" (Discussion section); "…Thus OT is causally involved in mediating beneficial effects of social experience on behavior and TL…" (Discussion section), and "The present data causally associates social stimulation with genetic and behavioral improvements as a function of endogenous OT" (subsection “Conclusion”). The authors did not measure behavior fitness, health or longevity.

Thanks to the reviewer for these comments. We have made some changes in the sentences in the manuscript as the reviewer requested. We hope that these changes address the reviewer’s concerns:

- Old sentence: “…The present rodent study demonstrates a key role of OT in socially mediated behavioral fitness and telomere length as an indicator of overall health and longevity…”.

New sentence: “…The present rodent study indicates that persistent social experience is causally associated with increased OT level, elevated exploratory behaviour, and telomere elongation….” (please see Discussion section).

- Old sentence: “…Thus OT is causally involved in mediating beneficial effects of social experience on behaviour and TL…".

New sentence: “…Thus OT is arguably involved in mediating beneficial effects of social experience on behaviour and TL…” (please see Discussion section).

- Old sentence: “…The present data causally associates social stimulation with genetic and behavioural improvements as a function of endogenous OT…".

New sentence: “…The present data causally associates social stimulation with genetic and behavioural improvements possibly as a function of endogenous OT…” (please see Discussion section).

Based on arguments presented in point 2 above, I don't think the results provided sufficient evidence to demonstrate a causal link between OT and TL. In order to demonstrate a causal link, at a minimum, the authors would need to demonstrate that in vivo OT administration can promote telomere length, and that the effect of social housing on telomere length are abolished in oxytocin knockout mice. Telomere length needs to be measuring in multiple tissues and not just ear punches. Also, no statements can be made regarding behavior fitness, health or longevity unless data is provided.

We agree with the reviewer that data in the present experiments cannot draw a direct line to connect changes in OT levels to TL, as the study was not originally designed for such questions. However, we still believe that the present data can make causal conclusion(s) for bridging the independent variable (i.e. social life) and dependent variables (i.e. enhanced novelty-seeking, increased levels of OT and/or telomere elongation). Please see our rationales in response to comment 2. Moreover, even though longevity was not measured in the present study, TL itself has been proposed as a potential biomarker for longevity in humans, mice, and other organisms in many studies (please, see Calado and Dumetrio, 2013, for example). More importantly, as shown by Monaghan’s team, TL in early life predicts lifespan (see Heidinger et al., 2012). From our viewpoint, results in our experiments significantly advance the understanding factors that may causally determine early life influences on TL.

5) Related to points 2 and 4 and given the discussion regarding how accurately oxytocin level in the plasma measured using RIA or ELISA reflects oxytocin level in the brain, the authors could measure oxytocin mRNA level or immunostain for oxytocin in brain sections, to further establish the correlation between oxytocin level and telomere length.

We appreciate this comment and apologize for our mistake. The authors confirm that 7 standard animals (4 males and 3 females) and 6 social animals (3 males and 3 females) from our previous experiment were included into the analysis of novelty-seeking behaviour in Experiment 1 in the present study. The inclusion of these animals has been conducted to increase sample size and to reduce within-group variation, thus achieving more precise estimates of average. To address the reviewer’s concern, we included a new sentence into the Material and Methods section which seemed more relevant that now reads:

“…Novelty-seeking behaviour analysis included 7 standard animals (4 males, 3 females) and 6 social animals (3 males, 3 females) from a previous experiment (Faraji et al., 2018) to increase sample size…”.

6) Results presented in Figure 2A-D are very similar to that presented in a previous publication from the same lab (Faraji et al., 2018). The authors should reference their previous results in the Results section, and not just the Materials and methods section.

Thank you for the great suggestion. Some of us (SR and RM) are analysing brain tissues for OT responses in the PVN and SON in the hypothalamus in response to social life, but unfortunately the results might not be reported in the present manuscript due to the extended amount of time required for such analyses. For the present study, based on previous findings in the literature we assumed that peripheral levels of OT can properly reflect central releases of OT in rats (for example, please see Wotjak et al., 1998). Furthermore, it seems that there is a coordinated axonal release of OT in the brain and in the circulatory system, and central OT and peripheral OT in animals can be released simultaneously. In humans, a correlation was shown between plasma and brain OT indicating that (at least under specific testing conditions) peripheral OT represents a suitable proxy of central OT (please see Lefvre and others 2017). Therefore, the authors believe that plasma OT concentration in the present study can be considered as a marker of its action in the brain. To address this concern, we have included a new statement in the Discussion section that now reads:

“…Axonal release of OT in the brain and in the circulatory system seem to be coordinated as central OT and peripheral OT in rats can be released simultaneously (Wotjak et al., 1998). In humans, a correlation was shown between plasma and brain OT indicating that peripheral OT represents a suitable proxy of central OT (Lefvre et al., 2017).

7) Statements such as "…Thus, social females' exploration in the corridor zone decreased -1.84 for each pM of OT concentration" (subsection “Novelty-Seeking Behaviour in Females was Significantly Influenced by Social Experiences”), and "…TL, therefore, only in social females increased 8.53 bp for each pM of OT level" (subsection “Novelty-Seeking Behaviour in Females was Significantly Influenced by Social Experiences”), do not make sense to me, as the data points for TL range from 4000 to 5700 bp, and that for OT range from around 40 to 125 pM. In light of huge differences in the levels of OT and TL, these precise interpretations do not make much sense to me.

We agree and are now convinced that the reviewer is absolutely right. Therefore, we omitted both sentences from the manuscript to address the reviewer’s concerns.

[Editors’ note: the author responses to the re-review follow.]

It is great that you did saline controls. Of course, it is a problem that none of the reviewers realized that. Please address this by making it far more evident in the manuscript. Also make sure that the saline groups make their way into the figures. Finally, please respond to the other issues raised by the reviewers. We look forward to your resubmission.

We appreciate this comment and have addressed these concerns by re-writing the corresponding sentence in the manuscript as the reviewer suggested. The new paragraph now reads:

“…Physiologic saline solution was injected subcutaneously in control groups (42-43 doses/rat) to avoid the confounding differential effect of stress resulted by repeated OT ANT injections.” (please, see Materials and methods section).

We also included the word “saline” in control group terminology in Figures 3 and 4 in order to highlight the saline injections in all control animals in Experiment 2 (please, see Figure 3A-D, and Figure 4A-D, F and H).